# Characterization of Rotavirus Infection in Hospitalized Children under 5 with Acute Gastroenteritis 5 Years after Introducing the Rotavirus Vaccines in South Korea

**DOI:** 10.3390/children9111633

**Published:** 2022-10-27

**Authors:** Hye Sun Yoon, Yong-Hak Sohn, Jeong Don Chae, Jiseun Lim, Seung Yeon Kim

**Affiliations:** 1Department of Pediatrics, Nowon Eulji Medical Center, School of Medicine, Eulji University, Seoul 01830, Korea; 2Seegene Medical Foundation, Seoul 04805, Korea; 3Department of Laboratory Medicine, Nowon Eulji Medical Center, School of Medicine, Eulji University, Seoul 01830, Korea; 4Department of Preventive Medicine, School of Medicine, Eulji University, Daejeon 34824, Korea

**Keywords:** rotavirus, vaccine, prevalence, genotype, vaccine effectiveness, siblings

## Abstract

We herein characterized rotavirus infection in hospitalized children under 5 years of age with gastroenteritis after introducing rotavirus vaccines in South Korea from 20 February 2012, to 31 March 2013. Enzyme-linked fluorescent immunoassay was performed to detect rotavirus antigens. G and P genotyping was performed using nested multiplex PCR. For the failed PCR samples, sequencing was conducted. We performed a test-negative case-control study to estimate vaccine effectiveness. Vaccine effectiveness was measured using a multivariate logistic regression model. Rotavirus was detected in 16 (13.2%) of the 121 patients, with a seasonal peak in April 2012. The dominant genotypes detected were G3P[8] (33.3%) and G4P[6] (26.7%), and vaccine effectiveness against rotavirus hospitalization was 84.9% [95% CI: 23.2–97.0] in the complete vaccinated group. A higher prevalence of rotavirus infection was observed among children with siblings than those without siblings (*p* < 0.001). Also, the presence of siblings was significantly associated with a history of nonvaccination (*p* < 0.001). In conclusion, the prevalence of rotavirus followed a decreasing trend, and there was no evidence of emergences of nonvaccine-type strains. Vaccine effectiveness against rotavirus hospitalization was 84.9%. Although children with siblings were more susceptible to rotavirus infection, they were less likely to receive vaccination against rotavirus.

## 1. Introduction

Rotavirus (RV) is a well-known pathogen that causes acute viral gastroenteritis and threatens the lives of children under the age of 5, causing 128,515 deaths (95% uncertainty interval [UI]: 105,138–155,133) worldwide each year [1]. The majority of children under 5 years of age worldwide are at risk for RV infection. Surface proteins G and P of RV determine its antigenicity and induce immunologically important type-specific neutralizing antibodies [2]. Since the type-specific neutralizing antibodies play an important role in defense against RV infection, many countries perform epidemiological surveillance of the genotypes of RV surface proteins G and P. The data obtained through epidemiological surveillance not only show the distribution pattern and epidemic trends of RV but also contribute to the development of RV vaccines because proteins G and P are the primary targets for vaccine development.

In 2006, two live oral vaccines, namely, RotaTeq (RV5, MSD) and Rotarix (RV1, GSK) vaccines, were introduced and made commercially available worldwide for infants. These vaccines were also recommended by WHO [3]. RV5 is a pentavalent vaccine that consists of a reassortment of G1, G2, G3, G4, and P1A[8] strains, a combination of human–bovine genes, whereas RV1 is a monovalent vaccine that consists of G1P[8], a single human strain [4]. The incidence of RV infection has decreased with the use of RV vaccines, preventing more than 28,000 deaths (95% UI, 14,600–46,700) in children under the age of 5 globally [5]. However, RV vaccine effectiveness is known to be lower in low- and middle-income countries than that in high-income countries [6,7]. Moreover, there is a concern that RV1 and RV5 could be less effective against RV strains not included in these vaccines [4]. In South Korea, RV5 was approved in 2007 and RV1 in 2008 by the Ministry of Food and Drug Safety, and RV5 and RV1 are applied at 2, 4, and 6 months of age and 2 and 4 months of age, respectively. However, RV vaccines have not yet been included in South Korea’s National Immunization Program (NIP) [8]. Although parents privately purchase RV vaccines for immunizing their children against RV infection, coverage of vaccination with RV vaccines is increasing continuously in South Korea.

In this prospective surveillance study, we aimed to investigate the prevalence, vaccine effectiveness (VE) against RV hospitalization, genotypes, and related factors of RV infection in hospitalized children under the age of 5 with acute gastroenteritis, excluding newborns, 5 years after the launching of RV vaccines at a single medical center in South Korea.

## 2. Materials and Methods

### 2.1. Study Participants

We collected stool samples from 127 children under the age of 5, excluding newborns less than 4 weeks of age, who were admitted to Daejeon Eulji University Hospital for acute gastroenteritis between 20 February 2012, and 31 March 2013. Inclusion criteria included the children with the following: (1) loose or watery stool three or more times over 12 h and (2) abnormal stools associated with fever, vomiting, or severe abdominal pain. Children with other diseases, such as pneumonia, otitis media, or other chronic gastrointestinal disorders, were excluded [9]. Clinical information of the participants was collected from the questionnaires completed by the parents or guardians of the children who agreed to provide information. The RV vaccination status of patients was determined based on parent reports and medical records. The enrolled patients included patients from both urban and rural areas, and no information on parental income was collected.

### 2.2. Research Method

#### 2.2.1. Rotavirus Antigen Detection

The RV antigen was detected using RIDASCREEN RV (R-Biopharm AG, Darmstadt, Germany) immunoassay or VIDAS rotavirus (bioMerieux Vitek, Marcy-l’Etoile, France). The detection rate of antigens in both test methods was equivalent [10].

#### 2.2.2. Specimen Preservation and Genotype Preanalysis

For RV genotyping, fecal samples were stored in a −70 °C freezer and thawed at approximately 20–22 °C (68–72 °F) before analysis. Next, 0.5–1.0 g of the fecal sample was resolved in 5 mL of 0.89% saline and centrifuged at 4000× *g*. Then genotyping was performed with 140 μL of the supernatant [11].

#### 2.2.3. Rotavirus Genotype Test

Multiplex polymerase chain reaction (PCR) was performed for RV genotyping. If the samples were not amplified successfully via multiplex PCR, Sanger’s sequencing was performed for genotyping as described previously [12,13]. In brief, RNA was pulled out from the fecal samples using the QIAamp viral RNA mini kit (Qiagen GmbH, Hilden, Germany). Then, the G and P genotypes were analyzed via reverse transcription polymerase chain reaction (RT-PCR) [12,13,14]. RT-PCR was used to identify the G genotypes with Beg9 and End9 primers. In addition, nested multiplex PCR was used to identify the G genotypes with End 9 and type-specific primers (aBT, aCT2, aFT3, aDT4, aAT8, and aFT9). To identify P genotypes, Con2 and Con3 primers were used in RT-PCR and Con3 and type-specific primers (1-T1, 2-T1, 3-T1, and 4-T1) in nested multiplex PCR according to the PCR conditions described previously [12,13]. Then, 10 μL of each PCR product was electrophoresed on 1.5% agarose gel (Sigma-Aldrich, St. Louis, MO, USA), stained with ethidium bromide, and observed under ultraviolet illumination. If the samples were not sufficiently amplified via nested multiplex PCR, Sanger’s sequencing was practiced with primary primers and primary RT-PCR products by Macrogen (Daejeon, Korea). Genotypes were identified by performing phylogenetic analysis with the Claustral Omega program (http://www.ebi.ac.uk/Tools/msa/clustalo/ (accessed on 10 June 2013)) of the European Bioinformatics Institute. RV reference strains for phylogenetic analysis were chosen as described by Matthijnssens et al. [11,15].

#### 2.2.4. VE against RV Hospitalization

We performed a test-negative case control study to estimate VE against RV hospitalization. The case group enrolled rotavirus-positive with hospitalized acute gastroenteritis and the control group enrolled rotavirus-negative with hospitalized acute gastroenteritis. The exposure odds ratio of vaccination among the case group and the control group has been used to estimate VE. VE with a 95% confidence interval was worked out with the following formula: VE = (1 − (minus sign) odds ratio) × 100%, where odds ratios comparing vaccinations between the case group and the control group were determined with a multivariate logistic regression model [16,17].

### 2.3. Statistical Analysis

Data on age, gender, birth weight, gestational age, method of delivery, presence of siblings, breastfeeding, and vaccination status were analyzed with SPSS 20.0 (IBM, Chicago, IL, USA). Pearson Chi-square test and Chi-square test for trend analysis were practiced. A *p*-value of <0.05 was considered statistically significant. Moreover, the VE of RV vaccines was estimated using a multivariate logistic regression model.

## 3. Results

### 3.1. Demographics and Prevalence of Rotavirus Infection

Of the 127 children recruited in this study, 6 were excluded from RV screening because of insufficient specimens. Of the remaining 121 patients, 16 patients (13.2%) were positive for the RV antigen (Table 1, Figure 1). Moreover, seasonal variation in RV infection was observed, with a peak in April 2012 (Figure 2). However, no statistically significant difference in RV positivity and demographic data, such as age, gender, gestation, birth weight, and delivery method, except the presence of siblings, was observed between the patients positive and negative for the RV antigen (Table 1).

### 3.2. Relationship between Rotavirus Infection and Vaccination and VE after Rotavirus Vaccination

In this study, two patients were excluded due to the uncertainty of vaccination. Among the remaining 119 participants, 65 (54.6%) had received complete vaccination, 12 (10.1%) had received incomplete vaccination (at least one dose), and 42 (35.3%) had not received any vaccination at all. Among the 77 participants who had received the RV vaccines, the type of vaccine was RV5 in 44 (57.1%), RV1 in 19 (24.7%), and unknown in 14 (18.2%). Moreover, the prevalence of RV infection was significantly higher in the nonvaccinated group than that in the complete vaccinated and incomplete vaccinated groups (3.1% in the complete vaccinated group, 16.7% in the incomplete vaccinated group, and 26.2% in the nonvaccinated group) (Table 2) (*p* = 0.002). Similarly, the multivariate logistic regression model revealed that the VE against RV hospitalization was 84.9% [95% CI: 23.2–97.0] in the complete vaccinated group adjusted for the presence of siblings and age. Interestingly, the VE against RV hospitalization was reported to be 91.9% [95% CI: 57.1–98.1] when it was not adjusted for the presence of siblings. In contrast, the multivariate logistic regression model revealed that the prevalence of RV infection was not decreased in the incomplete vaccinated group.

### 3.3. Genotypes of Rotavirus

Of the 121 fecal samples, the genotype in 1 of the 16 RV antigen-positive specimens could not be identified because the gene was not amplified. The remaining 15 cases were included for RV genotyping. The common genotypes identified were G3P[8] (5 patients, 33.3%) and G4P[6] (4 patients, 26.7%). The other genotypes identified were G1P[8] (2 patients, 13.3%), G3P[6] (1 patient, 6.7%), and G3P[4] (1 patient, 6.7%) (Figure 3). Four of the 16 patients positive for RV antigen had been vaccinated. Moreover, two patients each in the complete vaccinated and incomplete vaccinated groups had RV infection. Among these two RV-infected patients who had received complete vaccination, the genotype G2P[4] was detected in the one who had received the RV5 vaccine and the genotype G4P[6] was detected in another who had received the RV1 vaccine. Similarly, among the two patients who had received incomplete vaccination, the genotype G1P[8] was detected in the one who had received the RV5 vaccine and the genotype G3P[8] was detected in the patient in whom the type of vaccine was unknown.

### 3.4. Relationship between Rotavirus Infection and Breastfeeding and between Rotavirus Infection and the Presence of Siblings in Rotavirus Gastroenteritis Children with Hospitalization

Breastfeeding was defined when more than two-thirds of the daily intake was breast milk. No statistical significance was observed between breastfeeding and RV infection (*p* = 0.774) (Table 1). Besides, no statistical significance was observed between breastfeeding and the history of RV vaccination (*p* = 0.172) (Table 3). However, the prevalence of RV infection was significantly higher in children who had siblings than in those without siblings (*p* < 0.001) (Table 1). Similarly, the presence of siblings was significantly associated with the history of nonvaccination (*p* < 0.001) (Table 3).

## 4. Discussion

A systematic literature review by Burnett et al., revealed that the hospital admission rates have reduced by 59% and deaths from RV-induced diarrhea have reduced by 36% in the countries that have adopted RV vaccines [18]. We herein investigated the effect of RV vaccines in hospitalized children under the age of 5 with acute gastroenteritis at a single medical center 5 years after the introduction of RV vaccines.

A large-scale surveillance study in South Korea before the introduction of RV vaccines reported RV infection in 22% of all diarrhea cases with hospitalization [19]. Similar to the global data, the trend of RV infection in South Korea has been reported to decrease after RV vaccination [20]. Shim et al., reported that the RV prevalence in hospitalized children with gastroenteritis in South Korea before the vaccination was 25.9% in 2003–2006, which declined by 18.9% in 2007–2010 and by 15.8% in 2011–2015 after the vaccine introduction [21]. Similarly, the prevalence of RV infection in hospitalized children with gastroenteritis in the current study was 13.2%, although no previous comparable data were available at this site.

In temperate regions, outbreaks of RV infection mainly occur in cold, dry weather, whereas in tropical environments, less seasonality is observed [22]. Cho et al., reported that RV infections peaked in March and May during spring in South Korea [23]. In this study, a seasonal peak of RV infection was observed in April 2012, which was similar to the result of a study conducted prior to vaccine introduction [19].

In this study, VE against RV hospitalization was 84.9% in complete vaccinated children under 5 years of age, although it was not whole population-level data. VE rates of RV vaccine in children under 5 years of age have been reported to differ among high-, middle-, and low-income countries. In cases of severe RV infection, the VE of the RV vaccine is 80–90% in high-income countries and 40–60% in low- and middle-income countries [24,25,26]. Varied VE can be attributed to various factors, such as interference of high maternal antibody titers with vaccine strains, breastfeeding, malnutrition, coinfection, and altered intestinal microbiota. However, the exact mechanism underlying the variation in VE remains unclear [3,24]. Antibodies in breast milk are considered to interfere with RV vaccine efficacy and immunogenicity, resulting in low VE in low- and middle-income countries [3,27,28]. However, a recent study in New Delhi, India, observed no enhanced effect on VE when breastfeeding was stopped during the vaccination period [29]. Moreover, in 2021, WHO updated the position statement on RV vaccines and described that breastfeeding around the time of RV vaccination does not seem to significantly impair the response to RV vaccines [30]. The low VE of RV in low- and middle-income countries is challenging to combat the disease.

Breast milk has excellent nutritional components and provides several immunological benefits. Moreover, breastfeeding prevents RV infection [31,32,33]. Controversially, Shen et al., reported in a meta-analysis that no direct correlation was observed between RV infection and breastfeeding [34]. Similarly, no association between RV infection and breastfeeding was observed in hospitalized children with gastroenteritis in this study. Also, no association between breastfeeding and vaccination history was observed, suggesting that vaccination history significantly related to RV infection did not act as a bias for no relationship between RV infection and breastfeeding. Therefore, further studies on the relationship between RV infection and breastfeeding should be conducted.

After the introduction of RV vaccines, the VE was low for the strains not included in the RV vaccines [4]. Even a tiny difference in the strain-specific VE could cause selective change over time toward nonvaccine-type strains that can escape from the vaccine-induced immunity, declining the benefits of the RV vaccination [35]. In this study, the genotypes G3P[8], G4P[6], G2P[4], G1P[8], G3P[6], and G3P[4] were detected. G2P[4] and G4P[6] were detected in children completely vaccinated with RV5 and RV1, respectively. G2P[4] is a partially heterotypic strain of RV5, whereas G4P[6] is a fully heterotypic strain of RV1. Moreover, G1P[8], a homotypic strain of RV5, was detected in patients with incomplete RV5 vaccination. Therefore, there is a possibility that partially and fully heterotypic strains and incomplete vaccination may have influenced the decline in VE against RV hospitalization, despite the small sample size. Furthermore, although there was no evidence of the emergence of nonvaccine-type strains that can evade vaccine-induced immunity in this small sample, we encourage the surveillance of RV strains to track changes in dominant strains as in previous studies [36,37].

RV infection has been reported to be associated with the presence of siblings [38]; this finding was consistent with that of the present study. It seems that fecal-to-oral transmission plays a key role in the propagation of rotavirus within the family. Moreover, children with siblings were less likely to receive the rotavirus vaccine in the current study. Many studies have reported that herd immunity through children receiving RV vaccines protects their unvaccinated siblings [39,40,41,42,43]. Although rotavirus vaccination of children with siblings is more conducive to the development of herd immunity, the significant relationship between the presence of siblings and nonvaccine status is presumed to be due to the economic burden of self-paid RV vaccination, which is weighed by the presence of siblings. A previous study reported that the presence of siblings had a negative effect on self-paid vaccination [44]. Moreover, in this study, the VE against RV vaccination was higher when it was not adjusted for the presence of siblings than when it was adjusted. Therefore, to increase VE and spur the induction of RV herd immunity in South Korea, the NIP should include rotavirus vaccination, which can lower the economic burden on parents.

This study had some limitations. First, this study was a short-term and a single-center small-scale study. Second, it was conducted without considering other viral infections that cause enteritis. Third, the fecal RV antigen test may not be perfect, thus its results may have influenced the outcome of VE. The test-negative design used for VE in this study may be more susceptible than other designs to misclassification of disease outcomes due to incomplete diagnostic tests [45]. Fourth, we could not use the national immunization register to check the patient’s vaccination status because NIP did not include RV vaccination in South Korea. Fifth, participants excluded neonates less than 4 weeks of age; this might have influenced the prevalence of RV infection. Lastly, virus shedding due to vaccine and virus shedding due to infection could not be distinguished, although the present study was conducted in patients who were hospitalized due to RV symptoms. Nevertheless, this study is meaningful because it investigated the prevalence of RV infection as well as the VE against RV hospitalization and genotypes of RV in hospitalized children under 5 years of age with acute gastroenteritis 5 years after the introduction of RV vaccines in South Korea. Moreover, this study analyzed factors influencing RV infection, such as breastfeeding and the presence of siblings.

## 5. Conclusions

Five years after the introduction of the RV vaccine, the rate of RV prevalence in children under 5 years of age hospitalized for gastroenteritis was 13.2%, with a seasonal peak in April. The VE against RV hospitalization was 84.9% [95% CI: 23.2–97]. The genotypes of RV were G3P[8] (33.3%), G4P[6] (26.7%), G1P[8] (13.3%), G3P[6] (6.7%), and G3P[4] (6.7%), and there was no evidence of the emergence of strains of the nonvaccine type after the introduction of the vaccine. Breastfeeding, which is suspected as a factor related to RV infection, had no association with RV hospitalization. However, children with siblings, which is another factor related to RV infection, were found to be more susceptible to RV hospitalization than those without siblings. Moreover, such children were less likely to receive RV vaccines than those without siblings.

## Figures and Tables

**Figure 1 children-09-01633-f001:**
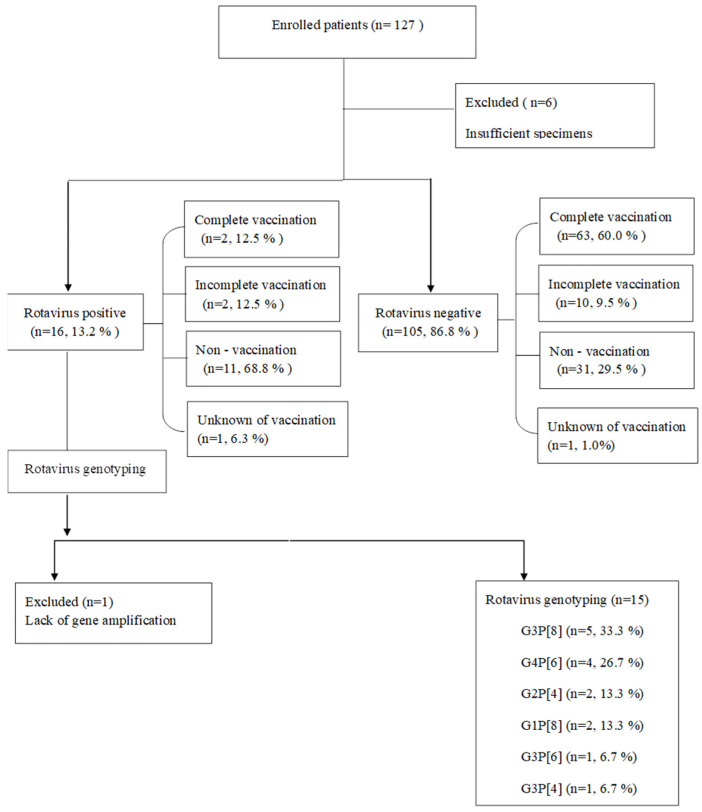
Flow diagram of the study population.

**Figure 2 children-09-01633-f002:**
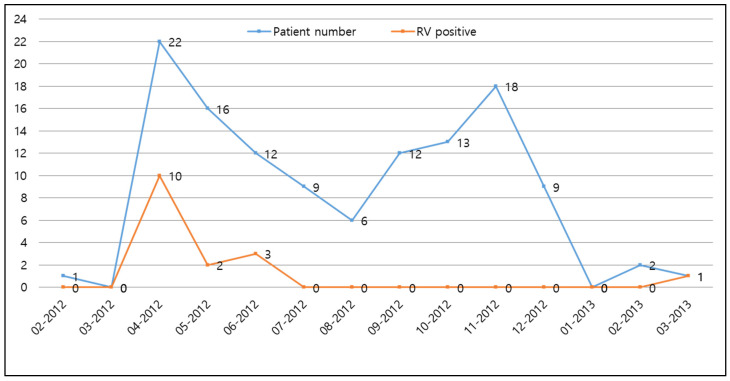
The monthly incidence of rotavirus infection in the pediatric ward from 20 February 2012, to 31 March 2013.

**Figure 3 children-09-01633-f003:**
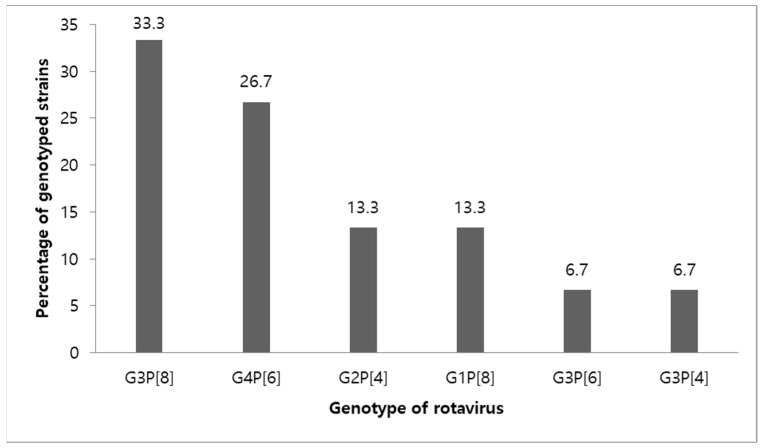
Genotype distribution in inpatient children with acute gastroenteritis less than 5 years ago.

**Table 1 children-09-01633-t001:** Demographic characteristics of rotavirus positive children (*n* = 16) and negative children (*n* = 105) with hospitalization by acute gastroenteritis.

Demographics	Number of Patients (%)		*p*-Value
	RV Ag-Positive	RV Ag-Negative	
age (months)			0.442
1–11 months	4 (7.7)	48 (92.3)	
12–23 months	5 (14.3)	30 (85.7)	
24–35 months	3 (18.7)	13 (81.3)	
36–47 months	3 (27.3)	8 (72.7)	
48–59 months	1 (14.3)	6 (85.7)	
gender			0.078
male	12 (18.2)	54 (81.8)	
female	4 (7.3)	51 (92.7)	
gestational age			0.224
≥37 weeks	16 (14.3)	96 (85.7)	
<37 weeks	0 (0.0)	9 (100)	
birth weight			0.327
>2500 g	16 (13.9)	99 (86.1)	
≤2500 g	0 (0.0)	6 (100)	
delivery			0.928
normal delivery	11 (13.4)	71 (86.6)	
cesarean section	5 (12.8)	34 (87.2)	
presence of sibling			<0.001 *
presence	14 (25.5)	41 (74.5)	
absence	2 (3.0)	64 (96.7)	
breastfeeding			0.774
yes	14 (13.6)	89 (86.4)	
no	2 (11.1)	16 (88.9)	
total	16 (13.2)	105 (86.8)	

Abbreviation: RV, rotavirus; * statistically significant.

**Table 2 children-09-01633-t002:** Relationship between vaccination history and rotavirus infection in hospitalized children with acute gastroenteritis.

	RV Vaccination Status			*p*-Value
	Complete V	Incomplete V	Non-V	
Number of patients (%)	65 (54.6)	12 (10.1)	42 (35.3)	
RV Ag				0.002 *
RV Ag-positive	2 (3.1)	2 (16.7)	11 (26.2)	
RV Ag-negative	63 (96.9)	10 (83.3)	31 (73.8)	

Abbreviation: RV, rotavirus; Ag, antigen; V, vaccination; * statistically significant.

**Table 3 children-09-01633-t003:** Relationship between vaccination history and breastfeeding, and the relationship between vaccination history and siblings in rotavirus gastroenteritis children with hospitalization.

	RV Vaccination Status			*p*-Value
	Complete V	Incomplete V	Non-V	
Number of Patient (%)	65 (54.6)	12 (10.1)	42 (35.3)	
breastfeeding				0.172
yes	56 (86.2)	12 (100)	33 (78.6)	
no	9 (13.8)	0 (0)	9 (21.4)	
presence of siblings				<0.001 *
presence	18 (27.7)	8 (66.7)	28 (66.7)	
absence	47 (72.3)	4 (33.3)	14 (33.3)	

Abbreviation: RV, rotavirus; V, vaccination; * statistically significant.

## Data Availability

The data presented in this study are available on request from the corresponding authors.

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
