# Peer review of "Characterization of Rotavirus Infection in Hospitalized Children under 5 with Acute Gastroenteritis 5 Years after Introducing the Rotavirus Vaccines in South Korea"

_children, 2022, doi:10.3390/children9111633_

Round 1

Reviewer 1 Report

This article provides information on the performance of oral rotavirus vaccines at a single site in South Korea. The study is appropriately designed and clearly presented, though case numbers are small. Recommend increased detail in the methodology section (see below) and minor edits to English language style in some sections.

Abstract: 

-       Line 20: add details re total number of rotavirus cases (not just proportion)

-       Line 22: clarify vaccine effectiveness measure ie ‘vaccine effectiveness against….’ (severe laboratory confirmed rotavirus gastroenteritis requiring hospitalisation?)

-       Line 26: suggest re-word ‘there was no evidence of emergence of non-vaccine type strains’

Introduction

-       Line 32/33: suggest re-write first sentence.

-       Line 36-41: reconsider sentence – epidemiological surveillance does not ‘control and prevent RV infection’ but it does show the distribution pattern and epidemic trends and contribute to the development of RV vaccines (second sentence is more accurate).

-       Line 46 – 48: clarify ‘global’ deaths

-       Line 53: suggest clarify immunisation ‘coverage’ is increasing

-       Line 53/54: suggest clarify that at the time of this study both RV1 and RV5 vaccines were available for private purchase? Consider adding details about the recommended age for administration RV1 & RV5 vaccine doses in South Korea.

Methods:

-       Study Participants: please provide more details about the study centre – urban vs rural? high income setting vs low- or middle- income setting?

-       Study Design: was this a prospective surveillance study or a retrospective review of medical notes/ laboratory results.

-       Consent Process: please clarify consent process; did study population include ALL children admitted to the hospital with gastroenteritis or only those whose parents or guardians agreed to provide information via questionnaires? 

-       Vaccine Status: how was vaccine status determined? Parental report? Medical records? Regional/ national mmunisation register?

Results: 

-       Table 1: consider providing more information about 1- 11 month age group; ie number age < 6 weeks (too young for rotavirus vaccination); number age < 6 – 8 months (RV positive samples in this age group could theoretically be vaccine virus shedding)

-       Table 3: purpose of Table 3 not clear; does period of breastfeeding mean ‘duration’ of breastfeeding? note duration of breastfeeding likely correlates to child’s age and coincident maturation of immune system? Ie complex interactions; consider remove Table 3;

-       Table 4: is sibling status reported for breastfeeding negative group only? (formatting not clear)

Discussion:

-       Line 174: opening sentence – suggest clarify if this is globally? Ie agross all income/ mortality settings

-       Line 185: please clarify no previous data available ‘at this site’.

-       Line 192 clarify ‘VE against …’  and age group < 5 years old.

-       Line 198 - 202: acknowledge that some studies in some settings have suggested that breast-feeding may impact VE, but suggest authors review/quote WHO 2021 updated position statement on rotavirus vaccines which says ‘Breastfeeding around the time of rotavirus vaccine administration does not seem to significantly impair the response to the rotavirus vaccines’ & clarify that next paragraph (204 – 211) is about RV infection, not vaccine effectiveness.

-       Line 221/222: study numbers are not large enough to make  conclusions about strain selection/ effect of VE, but could reword to say there was ‘no evidence of strain selection in this small sample’

-       Line 240: Limitations; suggest talk about sensitivity/ specificity of RV testing and effects on VE if faecal sample incorrectly reported as RV positive or RV negative; suggest discussion limitations re ascertainment of cases/ vaccination status; and choice of VE methodology…

-       Line 242; need to add ‘exclusion of newborns (and preferentially age ie < 4 weeks vs < 6 weeks” to exclusion criteria in Method section). 

-       Suggest clarify that all participants < 5 years had been eligible for rotavirus vaccination (ie born after June 2007/ January 2008?)

Conculusion
Line 250: please clarify ‘VE against ‘ Line 252; please clarify ‘no evidence of emerging strains’

Author Response

Thank you for your precious comments for taking the time to point out options to improve our paper.

We have revised the manuscript following your suggestion as follows.

This article provides information on the performance of oral rotavirus vaccines at a single site in South Korea. The study is appropriately designed and clearly presented, though case numbers are small. Recommend increased detail in the methodology section (see below) and minor edits to English language style in some sections.

Abstract: 

-       Line 20: add details re total number of rotavirus cases (not just proportion)

Answer: We’ve added the total number and highlighted it for you.

-       Line 22: clarify vaccine effectiveness measure ie ‘vaccine effectiveness against….’ (severe laboratory confirmed rotavirus gastroenteritis requiring hospitalisation?)

Answer: We’ve clarified as ‘the vaccine effectiveness against severe rotavirus infection’ and highlighted it for you.

-       Line 26: suggest re-word ‘there was no evidence of emergence of non-vaccine type strains’

Answer: We’ve reworded as your suggestion and highlighted for you.

Introduction

-       Line 32/33: suggest re-write first sentence.

Answer: We’ve rewritten the first sentence and highlighted it for you.

-       Line 36-41: reconsider sentence – epidemiological surveillance does not ‘control and prevent RV infection’ but it does show the distribution pattern and epidemic trends and contribute to the development of RV vaccines (second sentence is more accurate).

Answer: We’ve amended the sentence and highlighted it for you.

-       Line 46 – 48: clarify ‘global’ deaths

Answer: We’ve clarified ‘global’ deaths and highlighted it for you.

-       Line 53: suggest clarify immunisation ‘coverage’ is increasing

Answer: We’ve clarified the sentence and highlighted it for you.

-       Line 53/54: suggest clarify that at the time of this study both RV1 and RV5 vaccines were available for private purchase? Consider adding details about the recommended age for administration RV1 & RV5 vaccine doses in South Korea.

Answer: RV1 and RV5 vaccines were approved by the Ministry of Food and Drug Safety of South Korea in June 2007 and January 2008, respectively. Since then, parents or guardians could immunize their child by private purchase. We’ve highlighted the sentence for you.

We’ve added details about the recommended age for administration RV1& RV5 vaccine doses in South Korea.

Methods:

-       Study Participants: please provide more details about the study centre – urban vs rural? high income setting vs low- or middle- income setting?

Answer: Enrolled patients were mixed urban and rural residents and no information on parental income was collected in this study. We’ve added this information in the part of 2.1 Study participants and highlighted it for you. 

-       Study Design: was this a prospective surveillance study or a retrospective review of medical notes/ laboratory results.

Answer: This study is a prospective surveillance study. We’ve added the phrase in the last sentence in the introduction and highlighted it for you. 

-       Consent Process: please clarify consent process; did study population include ALL children admitted to the hospital with gastroenteritis or only those whose parents or guardians agreed to provide information via questionnaires? 

Answer: We’ve clarified the consent process and highlighted it for you.

-       Vaccine Status: how was vaccine status determined? Parental report? Medical records? Regional/ national mmunisation register?

Answer: We’ve clarified how the information of vaccine status was obtained and highlighted it for you.

Results: 

-       Table 1: consider providing more information about 1- 11 month age group; ie number age < 6 weeks (too young for rotavirus vaccination); number age < 6 – 8 months (RV positive samples in this age group could theoretically be vaccine virus shedding)

Answer: I am sorry that we didn’t have more information about the 1-11month age group. But we’ve added your concern about vaccine virus shedding in the limitation part and highlighted it for you.

-       Table 3: purpose of Table 3 not clear; does period of breastfeeding mean ‘duration’ of breastfeeding? note duration of breastfeeding likely correlates to child’s age and coincident maturation of immune system? Ie complex interactions; consider remove Table 3;

Answer: We’ve removed table 3 based on your comment.

-       Table 4: is sibling status reported for breastfeeding negative group only? (formatting not clear)

 Answer: We’ve changed formatting Table 4 and highlighted it for you. 

Discussion:

-       Line 174: opening sentence – suggest clarify if this is globally? Ie agross all income/ mortality settings

Answer: We’ve clarified the opening sentence and highlighted it for you.

-       Line 185: please clarify no previous data available ‘at this site’.

Answer: We’ve added the ‘at this site’ and highlighted it for you.

-       Line 192 clarify ‘VE against …’  and age group < 5 years old.

Answer: We’ve clarified the sentence and highlighted it for you.

-       Line 198 - 202: acknowledge that some studies in some settings have suggested that breast-feeding may impact VE, but suggest authors review/quote WHO 2021 updated position statement on rotavirus vaccines which says ‘Breastfeeding around the time of rotavirus vaccine administration does not seem to significantly impair the response to the rotavirus vaccines’ & clarify that next paragraph (204 – 211) is about RV infection, not vaccine effectiveness.

Answer: We’ve quoted the WHO 2021 updated position statement on rotavirus vaccines and highlighted it for you.

We’ve clarified the next paragraph and highlighted it for you.  

-       Line 221/222: study numbers are not large enough to make  conclusions about strain selection/ effect of VE, but could reword to say there was ‘no evidence of strain selection in this small sample’

Answer : We’ve reworded based on your comment and highlighted it for you.

-       Line 240: Limitations; suggest talk about sensitivity/ specificity of RV testing and effects on VE if faecal sample incorrectly reported as RV positive or RV negative; suggest discussion limitations re ascertainment of cases/ vaccination status; and choice of VE methodology…

Answer: We’ve added the contents in limitation part and highlighted it for you.

-       Line 242; need to add ‘exclusion of newborns (and preferentially age ie < 4 weeks vs < 6 weeks” to exclusion criteria in Method section). 

Answer: We’ve added the contents in the limitation part and highlighted it for you.

-       Suggest clarify that all participants < 5 years had been eligible for rotavirus vaccination (ie born after June 2007/ January 2008?)

Answer: Thank you for your suggestion.

Conculusion
Line 250: please clarify ‘VE against ‘ Line 252; please clarify ‘no evidence of emerging strains’

Answer: We’ve clarified the phrase based on your comments and highlighted them for you.

Reviewer 2 Report

The manuscript is clear but needs the approval of ethical committee. In addition, in the results section, the statistically significance between the presence of sibling and RV positivity should be highlighted in line 114. Otherwise, this variable should be omitted from Table 1 and analyzed separately.  

Author Response

We deeply appreciate your valuable comments and opinions.

The manuscript is clear but needs the approval of ethical committee. In addition, in the results section, the statistically significance between the presence of sibling and RV positivity should be highlighted in line 114. Otherwise, this variable should be omitted from Table 1 and analyzed separately.  

Answer : This study protocol was approved by Eulji University Hospital Institutional Review committee on February 10, 2012 (12-64). We’ve highlighted the sentence for you.

We’ve added the phrase ‘except the presence of siblings’ in the results section and highlighted it for you.  
